# Oxymatrine Attenuates Tumor Growth and Deactivates STAT5 Signaling in a Lung Cancer Xenograft Model

**DOI:** 10.3390/cancers11010049

**Published:** 2019-01-07

**Authors:** Young Yun Jung, Muthu K. Shanmugam, Acharan S. Narula, Chulwon Kim, Jong Hyun Lee, Ojas A. Namjoshi, Bruce E. Blough, Gautam Sethi, Kwang Seok Ahn

**Affiliations:** 1Department of Science in Korean Medicine, Kyung Hee University, 24 Kyungheedae-ro, Dongdaemun-gu, Seoul 02447, Korea; ve449@naver.com (Y.Y.J.); sunny10526@nate.com (C.K.); mirue88@nate.com (J.H.L.); 2Department of Pharmacology, Yong Loo Lin School of Medicine, National University of Singapore, Singapore 117600, Singapore; phcsmk@nus.edu.sg; 3Naula Research, Chapel Hill, NC 27516, USA; anarula1@nc.rr.com; 4Comorbidity Research Institute, College of Korean Medicine, Kyung Hee University, 24 Kyungheedae-ro, Dongdaemun-gu, Seoul 02447, Korea; 5Center for Drug Discovery, RTI International, Research Triangle Park, Durham, NC 27616, USA; onamjoshi@rti.org (O.A.N.); beb@rti.org (B.E.B.); 6Department of Korean Pathology, College of Korean Medicine, Kyung Hee University, 24 Kyungheedae-ro, Dongdaemun-gu, Seoul 02447, Korea

**Keywords:** oxymatrine, STAT5, apoptosis, NSCLC

## Abstract

Oxymatrine (OMT) is a major alkaloid found in radix *Sophorae flavescentis* extract and has been reported to exhibit various pharmacological activities. We elucidated the detailed molecular mechanism(s) underlying the therapeutic actions of OMT in non-small cell lung cancer (NSCLC) cells and a xenograft mouse model. Because the STAT5 signaling cascade has a significant role in regulating cell proliferation and survival in tumor cells, we hypothesized that OMT may disrupt this signaling cascade to exert its anticancer effects. We found that OMT can inhibit the constitutive activation of STAT5 by suppressing the activation of JAK1/2 and c-Src, nuclear localization, as well as STAT5 binding to DNA in A549 cells and abrogated IL-6-induced STAT5 phosphorylation in H1299 cells. We also report that a sub-optimal concentration of OMT when used in combination with a low dose of paclitaxel produced significant anti-cancer effects by inhibiting cell proliferation and causing substantial apoptosis. In a preclinical lung cancer mouse model, OMT when used in combination with paclitaxel produced a significant reduction in tumor volume. These results suggest that OMT in combination with paclitaxel can cause an attenuation of lung cancer growth both in vitro and in vivo.

## 1. Introduction

Non-small cell lung cancer (NSCLC) is the most common and lethal cancer worldwide, affecting both men and women [1,2]. One of the most frequently encountered causes for NSCLC has been found to be tobacco smoking [3]. NSCLC is more common in women and in East Asia is often associated with environmental factors such as exposure to occupational carcinogens, second-hand smoke, pollution, and genetic susceptibility [2,4,5,6,7,8,9]. Several common variant allele frequencies of alterations have been also found in NSCLC patients such as somatic mutations, homozygous deletions, focal amplifications, and significant up- or downregulation of gene expression in Kirsten rat sarcoma (KRAS) and epidermal growth factor receptor (EGFR) [2]. Current treatments for advanced lung cancer include chemotherapy, targeted therapies, and surgical removal of tumor followed by radiation therapy [10]. However, the existing treatment modalities exhibit serious side effects in patients. Therefore, the identification and development of novel and efficient drugs from alternative sources such as Mother Nature are urgently required [11,12,13,14,15,16,17,18,19,20,21].

Signal transducer and activator of transcription 5 (STAT5) is a pro-survival transcription factor that is known to regulate several critical cellular functions and in the maintenance of cellular homeostasis and is often found to be deregulated in cancers [22,23,24]. Among the STAT family of proteins, STAT5 has been closely implicated in the development of a variety of cancers including hematological malignancies and solid tumors [25,26]. Constitutive phosphorylation of tyrosine residue (Tyr694/Tyr699) and transcriptional activation of STAT5 have been observed in diverse malignancies [27,28,29,30]. Phosphorylation of tyrosine residue occurs through the activation of intracellular kinases such as JAK1/2 and c-Src [31,32,33]. Moreover, potential mechanistic crosstalk of STAT5 with the PI3K/AKT pathway has also been observed during normal mammary gland development and during tumorigenesis [34,35,36]. In A549 lung cancer cells, epidermal-growth-factor-induced COX-2 expression has been implicated in tumor progression through the activation of the STAT5 signaling pathway [37]. It has also been found that the silencing of miR10a reversed resistance lung cancer resistance to cisplatin by modulating the TGFβ/Smad2/STAT3/STAT5 pathway [38]. Several studies have also reported that natural and/or synthetic pharmacological agents that can modulate STAT5 signaling pathway may have tremendous potential in the therapy of cancer [39,40,41,42,43,44]. STAT5 can play a pivotal role in the development of Tregs and may also be associated with suppression of antitumor immunity and thus can serve as a potential therapeutic target [31,45].

Traditional Chinese medicine (TCM) originated in China and it has long been practiced in several East Asian countries for over thousands of years [46]. In view of its important contribution to Chinese medical practices, the Chinese state food and drug administration has approved TCM for the clinical treatment of solid tumors [46,47]. TCM has been used as an adjuvant therapy in patients undergoing standard chemotherapy and radiotherapy and has been shown to reduce the adverse effects associated with cancer therapy [48]. Furthermore, clinical trials in cancer patients with TCM formulations were found to reduce tumor burden, strengthen immune function, and improve overall quality of life [48,49]. Matrine and oxymatrine (OMT) are two major alkaloids found in radix *Sophorae flavescentis* belonging to the family Leguminosae that can exhibit diverse pharmacological activities such as anti-inflammatory, anti-viral, anti-allergic, anti-cancer, and cardiovascular protective effects [46,50].

Interestingly, OMT was found to abrogate breast cancer cell proliferation and downregulate the Wnt/β-catenin signaling pathway [51]. OMT also inhibited the growth of PANC-1 pancreatic cancer cells and induced apoptosis by downregulating anti-apoptotic protein such as Bcl-2 and the induction of caspase 3 [52]. OMT either alone or in combination with angiogenesis inhibitor NM3 synergistically inhibited the growth of human gastric cancer cells in vitro and abrogated the growth of SGC-7901 cells in vivo [53]. In another study, OMT was noted to attenuate the growth, induce apoptosis, and inhibit the expression of Bcl-2 protein with a concomitant increase in the expression of the *p53* gene in human hepatoma SMMC-7721 cells in vitro [54]. In another study, using human hepatocellular carcinoma cells HepG2 and SMMC-7721, OMT reduced proliferation in a dose-dependent manner and induced apoptosis. Moreover, in combination with 5-fluorouracil, OMT can produce a synergistic anti-tumor effect both in vitro and in vivo [55]. Several recent studies have demonstrated the anticancer effects of OMT in diverse cancer cell lines such as prostate cancer [56], ovarian cancer [57], gastric cancer [58], colorectal cancer [59,60,61], breast cancer [62,63], bladder cancer [64], hepatocellular carcinoma [55], esophageal carcinoma [65], osteosarcoma [66,67], cervical cancer [68,69], gallbladder carcinoma [70], laryngeal carcinoma [71], hemangioma [72], lung cancer [73,74,75,76], synovial sarcoma [77], glioblastoma [78,79], and nasopharyngeal carcinoma [80]. The molecular mechanism(s) of action of OMT was found to be mediated by inducing cell cycle arrest and apoptosis and by causing an inhibition of angiogenesis and metastasis [57,64,81,82].

In addition, matrine has also been shown to inhibit the growth of several organ-specific cancers such as breast cancer, gastric cancer, gallbladder cancer, osteosarcoma, and hepatocellular carcinoma by modulating pro-survival cell signaling pathways and the induction of apoptosis [47]. In breast cancer cells, matrine suppressed the phosphorylation of NF-κB and its subsequent nuclear translocation in MCF-7 and BT549, MDA-MB-231 triple negative breast cancer cells [83]. In another study, matrine was found to induce cell cycle arrest and apoptosis by suppressing the expression of micro-RNA21, upregulating the expression of tumor suppressor protein PTEN and thereby inhibiting the PI3K/AKT signaling pathway [84]. In gastric cancer cells, matrine induced dose- and time-dependent apoptosis that was found to be associated with an increase in caspase-3 activity [85]. Similarly, in MKN45 gastric cancer cells, matrine inhibited proliferation, upregulated caspase-3 and -7, and induced apoptosis [86]. Dysregulation of microRNAs, a class of small, non-coding, regulatory RNA molecules involved in gene expression, has been reported to be strongly associated with cancer initiation and progression. Interestingly, matrine can alter microRNA expression profiles in SGC-7901 human gastric cancer cells. Matrine upregulated 128 miRNAs substantially exhibiting >2-fold expression changes in treated cells compared to the untreated control cells [87].

In this study, we primarily focused to investigate the potential anticancer effects of OMT in NSCLC cell lines and a xenograft mouse model. We found that the anti-neoplastic effects of OMT may be primarily mediated through the attenuation of the STAT5 signaling axis. Additionally, OMT was found to abrogate STAT5 activation through multiple mechanisms(s), whereas matrine exhibited a minimal effect on the STAT5 signaling cascade.

## 2. Results

### 2.1. OMT Suppresses Constitutive STAT5 Phosphorylation in NSCLC Cells

Several previous studies have shown that STAT5 plays a significant role in regulating tumor survival and proliferation [22,31,88,89,90]. We tested whether OMT can regulate constitutive STAT5 activation on A549 lung cancer cells. A549 cells were treated with various indicated concentrations for 6 h or time intervals with 200 μM OMT. As shown on Figure 1B,C, constitutive phosphorylation of STAT5 was markedly suppressed upon OMT exposure.

### 2.2. OMT Inhibits STAT5 DNA Binding and Nuclear Translocation

Because STAT5 dimerization is accompanied by subsequent translocation into the nucleus and the transcription of target genes, we examined whether OMT can inhibit STAT5 binding to DNA by EMSA. The result shows that OMT can indeed inhibit STAT5-DNA binding (Figure 1D) in a dose- and time-dependent fashion. Dimerized STAT5 can translocate into the nucleus to regulate gene transcription. Therefore, we examined whether OMT can suppress STAT5 translocation by immunocytochemistry. The results show that the translocation of STAT5 into nucleus was modulated by OMT (Figure 1E).

### 2.3. OMT Suppresses the Phosphorylation of Signaling Kinases

Because STAT5 has been activated by upstream signaling kinases, we examined that OMT can also suppress the phosphorylation of JAK1, JAK2, and Src. Cells were treated, and OMT suppressed phospho-JAK1, phospho-JAK2, and phospho-Src, both in dose- and time-dependent manners (Figure 1F,G).

### 2.4. OMT Abrogates the Activation of IL-6-Induced STAT5 Phosphorylation and the Activation of Upstream Kinases

We also examined whether OMT also has a modulatory effect on the inducible STAT5, and upstream signaling kinases in H1299 cells. Cells were pretreated with OMT, and STAT5 phosphorylation was then induced by IL-6 treatment. As shown on Figure 2A–C, OMT clearly suppressed phosphorylation of STAT5 and attenuated JAK1, JAK2, and Src activation.

### 2.5. OMT Exhibits a Substantial Inhibitory Effect on the Phosphorylation of STAT5 as Compared to Matrine

OMT and matrine have very similar chemical structures, but matrine has one less oxygen. We next determined the effect of martine on phospho-STAT5 expression in NSCLC cells. As shown in Figure 2D,E, OMT exhibited substantial suppressive effects on constitutive and inducible phospho-STAT5 expression in A549 and H1299 cells, respectively.

### 2.6. OMT Suppresses the Expression of Various Oncogenic Proteins and Induces Cell Death in NSCLC Cells

STAT5 activation can regulate various oncogenic proteins involved in cell proliferation, cell survival, and angiogenesis. Western blot analysis indicates that OMT-treated cells showed a reduction in the levels of Bcl-2, Bcl-xl, Survivin, IAP-1, IAP-2, COX-2, VEGF, and MMP-9 proteins (Figure 3A). Furthermore, anti-apoptotic protein expression was also suppressed at their mRNA levels upon OMT exposure (Figure 3B). We also found that caspase-3 and PARP cleavage was induced upon the OMT treatment in A549 cells (Figure 3C). As shown in Figure 3D, OMT-treated cells were arrested in the Sub G1 and G0/G1 phase. The Sub G1 phase population was 7% in non-treated cells and then increased to 12 and 21% in cells treated with 100 and 200 μM OMT. Additionally, both A549 cells and H1299 cells showed positive staining for Annexin V upon 200 μM OMT treatment (Figure 3E,F). To analyze the effect on cell growth, an MTT assay was used and the growth potential of both A549 and H1299 cells was found to be significantly abrogated (Figure 3G).

### 2.7. Combination Treatment of OMT and Paclitaxel Exhibits Significant Anti-Cancer Effects

We carried out cytotoxicity analysis and determined optimal concentrations using Calcusynsoftware (BIOSOFT, Ferguson, MO). A549 cells were incubated with a range of concentrations of OMT as well as paclitaxel and cell viability was determined by MTT assay. When incubated for 24 h, OMT and paclitaxel in combination exerted significant cytotoxicity in a concentration-dependent manner (Figure 4A). We found that the combination of 100 μM OMT and 10 nM paclitaxel exerted a substantial potentiation effect (CI = 0.742). For this reason, the combination of 100 μM OMT and 10 nM paclitaxel was used further to determine the possible mechanism of chemosensitazation of NSCLC cells to paclitaxel upon OMT treatment. As shown in Figure 4B, OMT and paclitaxel co-treatment induced significant apoptosis as compared to the single treatment. We then analyzed the activation of STAT5 and upstream signaling kinases via Western blot analysis. The results indicated that STAT5 and upstream signaling kinases JAK1, JAK2, and Src phosphorylation was also substantially reduced by OMT and paclitaxel treatment (Figure 4C,D). The effect of drug combination on apoptosis was analyzed next. The results indicated that OMT and paclitaxel co-treatment substantially enhanced apoptosis, as observed in multiple assays (Figure 5A,B). Furthermore, we found that the expression of various oncogenic proteins was substantially suppressed by OMT and paclitaxel co-treatment (Figure 5C,D). The result also indicated that OMT and paclitaxel co-treatment induced a more effective increase in caspase-3 and PARP cleavage as compared to the individual agents (Figure 5D).

### 2.8. OMT Induces Antitumor Effects in a Xenograft Mouse Model and Modulates STAT5 Activation in Tumor Tissues

We examined the pharmacological potential of OMT on the growth of subcutaneously implanted human lung cancer cells A549 in nude mice. The experimental protocol is specified in Figure 6A. In Group I, as the control group, tumor volume sharply increased. However, Groups II, III, and IV had only a marginal increase in tumor volume (Figure 6C). On Day 25, animals were sacrificed, and final tumor size and tumor weight were measured. Tumor size and weight were significantly decreased in Groups II, III, and IV as compared to the control group (Group I) (Figure 6B,D). However, OMT and paclitaxel treatment had no effects on the body weight of mice (Figure 6E). We also analyzed the effect of OMT and paclitaxel co-treatment on A549 human lung tumor tissues by immunohistochemical analysis and found that co-treatment of OMT reduced phospho-STAT5 expression. Moreover, the data showed that OMT also downregulated the expression of Ki-67 protein, which is a biomarker of proliferation (Figure 7A). Additionally, we investigated whether OMT and paclitaxel can regulate STAT5 expression in A549 tumor tissues. Interestingly, OMT and paclitaxel were found to significantly decrease the expression level of phospho-STAT5, but there was no change in the levels of total STAT5 protein (Figure 7B). In order to determine whether OMT can suppress various proteins related with anti-apoptosis, proliferation, and angiogenesis, we carried out Western blot analysis using tumor tissues proteins. The results indicate that Bcl-2, Bcl-xl, Survivin, IAP-1, IAP-2, COX-2, VEGF, and MMP-9 protein levels were significantly reduced upon OMT and paclitaxel treatment (Figure 7C). Additionally, OMT was found to induce substantial expression of caspase-3 and cause PARP cleavage (Figure 7D).

## 3. Discussion

The goal of this study was primarily to determine the anti-cancer effects of oxymatrine in NSCLC cells and decipher its underlying molecular mechanism(s) of action. STAT families of proteins are often found to be deregulated in lung cancer cells [31,88,89,90]. In addition to the detection of hyperactivated STAT3 transcription factor in cancer cells, other studies have highlighted the importance of another member of the STAT family, namely STAT5, which has been reported to be frequently activated by various tyrosine kinases, oxidant stress, and ROS metabolism in cancer [91]. In prostate cancer cells, STAT5 acts as a marker for disease recurrences, and detection of high nuclear phosphorylated STAT5 is an indicator of both early recurrence and shorter overall survival [92]. Interestingly, in 71 NSCLC patients, immunoexpression analysis revealed significantly higher STAT5 levels in pT2 tumors and a positive correlation between STAT5 and COX-2 levels were also observed [88,89]. Moreover, immunohistochemical analysis for the expression of STAT5 was carried out in 92 NSCLC samples. It was noted that STAT5 was found to be overexpressed in 41.3% in the cytoplasm and 32.6% in the nucleus and was correlated with Bcl-xL overexpression [90].

Natural and synthetic compounds have been shown to inhibit upstream kinases involved in the STAT5 signaling pathway. For example, a small molecular weight compound containing a phosphotyrosyl-mimicking salicylic acid group has been shown to efficiently bind to the SH2 domain of STAT5 and inhibit STAT5-SH2 domain protein interactions in leukemic cells [93]. There are also several molecules that have been shown to inhibit STAT5 phosphorylation and inhibit leukemia cell proliferation [94,95,96,97]. Some recent studies have also reported that natural pharmacological agents such as formononetin can modulate the STAT5 signaling pathway in multiple myeloma cells, and these compounds may have tremendous potential in the prevention and therapy of cancer [39,40,41,42,43,44]. Indeed, the discovery of several novel small-molecule drugs that target JAK-STAT5 are in progress, although at present no specific STAT5 inhibitors are available for testing in clinical trials. It is well known that IL-6 can activate the early acute phase response by activating STAT3 (Tyr705) and STAT5 (Tyr694/Tyr699) signaling pathways and plays an important role in diseases associated with chronic inflammation, including cancer development [31,98,99,100]. It has been previously reported that OMT can exhibit significant anti-cancer effects on a variety of tumor cells by modulation of multiple signaling pathways. In this study, we report for the first time that OMT exhibits potent anti-cancer activity by abrogating the STAT5 signaling pathway in NSCLC cells. We demonstrate that OMT suppressed the constitutive and IL-6 induced phosphorylation of STAT5 in A549 and H1299 cells along with the concomitant suppression of JAKs and c-Src kinases. Interestingly, we noted that matrine, which is quite identical to OMT in its chemical structure, exhibited its inhibitory effect on STAT5 phosphorylation to a lesser extent as compared to OMT. The OMT-mediated modulation of the STAT5 signaling axis was associated with the inhibition of the proliferation and induction of apoptosis in NSCLC cells. Therefore, the findings suggest that OMT could be an effective and novel inhibitor of STAT5 phosphorylation, DNA binding, and subsequent nuclear translocation in NSCLC cells. The underlying molecular mechanism(s) by which OMT can affect STAT5 activation was also investigated in detail. STAT5 can interact with JAK1/2 kinases as a scaffold, and this interaction often leads to STAT5 phosphorylation at Tyr694/Tyr699 residues. Besides attenuating JAK1/2 activation, OMT also inhibited c-Src phosphorylation involved in STAT5 activation in tumor cells. Therefore, we can conclude that OMT can regulate STAT5 activation by repressing the phosphorylation of JAK1/2 and Src proteins.

We also report that sub-optimal concentration of OMT when used in combination with a low dose of mitotic inhibitor paclitaxel produced significant anti-cancer effects in NSCLC cells by abrogating proliferation and causing downregulation in the expression of various oncogenic proteins. In addition, paclitaxel was found to induce phosphorylation of STAT5 and its upstream kinases. OMT also dose-dependently inhibited paclitaxel-induced phosphorylation of STAT5, JAK1/2, and activated caspase 3. Our findings also clearly show that OMT can negatively regulate STAT5 protein, resulting in the inhibition or proliferation and augmentation of apoptosis induced by chemotherapy in lung cancer cells. In a recent study, Liu et al. reported that OMT enhanced the inhibitory effect of 5-fluorouracil on hepatocellular carcinoma both in vitro and in vivo [55]. A similar study reported by the same group showed that OMT also augmented the anti-tumor activity of oxaliplatin in colon cancer cells [60]. In our preclinical study, OMT at a dose of 30 mg/kg when administered intraperitoneally (i.p.) thrice a week for 20 days did not induce a significant change in lung tumor volume. However, when used in combination with paclitaxel, the combination group produced a significant reduction in tumor volume. Overall, our results suggest that OMT in combination with paclitaxel can significantly inhibit lung cancer cell growth both in vitro and in vivo. Similarly, in a recent study, Wei Li et al. showed that OMT can inhibit the activation of both wild-type and mutant epidermal growth factor receptor in NSCLC. Moreover, OMT prominently suppressed tumor growth in a xenograft mouse model [76]. Thus, OMT appears to function as a novel therapeutic agent that can be effectively employed for the treatment of various malignancies.

## 4. Materials and Methods

### 4.1. Reagents

Oxymatrine (OMT, Figure 1A) was purchased from Cayman Chemical (Michigan, USA) and matrine was purchased from Sigma-Aldrich (St. Louis, MO, USA). OMT was stored in a 100 mM stock solution with dimethyl sulfoxide at −20 °C, diluted in the cultured media for in vitro or PBS in vivo experiments. 3-(4,5-Dimethylthiazol-2-yl)-2,5-diphenyltetrazolium bromide (MTT), propidium iodide (PI), Tris base, glycine, NaCl, sodium dodecylsulfate (SDS), and bovine serum albumin (BSA) were purchased from Sigma-Aldrich (St. Louis, MO, USA). LightShift^®^ Chemiluminescent EMSA kit were purchased from Thermo Fisher Scientific Inc (Waltham, MA, USA). Alexa Fluor^®^ 488 donkey anti-rabbit IgG (H+L) antibody was obtained from Life Technologies (Grand Island, NY, USA). Whole-cell lysates of tumor tissues were obtained with T-PER Tissues Protein Extraction Reagent (Pierce, Rockford, IL, USA).

### 4.2. Cell Lines and Culture Conditions

Human non-small cell lung carcinoma (NSCLS) cells A549 and H1299 cells were obtained from American Type Culture Collection (Manassas, VA, USA). A549 cells were cultured with DMEM-low medium containing 10% inactivated fetal bovine serum (FBS) and 1% penicillin-streptomycin. Cells were incubated at 37 °C in 5% CO_2_ conditions.

### 4.3. Western Blot Analysis

Western blot analysis was performed as described previously [101].

### 4.4. Electrophoretic Mobility Shift Assay (EMSA) for STAT5-DNA Binding

STAT5-DNA binding was analyzed by an electrophoretic mobility shift assay (EMSA) as described previously [102].

### 4.5. Immunocytochemistry for STAT5 Localization

Immunocytochemistry was performed as described previously [103].

### 4.6. Reverse Transcription Polymerase Chain Reaction (RT-PCR) for RNA Analysis

A549 cells were treated with various concentrations of OMT (0, 25, 50, 100, 200 μM) for 24 h. Cells were harvested, suspended in trizol, and then incubated with chloroform and isopropanol. Extracted RNA was reverse-transcribed into cDNA and examined by RT-PCR using superscript reverse transcriptase and Taq polymerase (TAKARA, Tokyo, Japan). RT-PCR was performed with Bcl-2 at 94 °C for 2 min, 94 °C for 15 s, 56 °C for 30 s, and 72 °C for 1 min, with 30 cycles and an extension at 72 °C for 5 min. Bcl-xl was performed at 94 °C for 2 min, 94 °C for 30 s, 56 °C for 30 s, and 72 °C for 1 min, with 25 cycles and an extension at 72 °C for 7 min. Survivin was performed at 94 °C for 5 min, 94 °C for 30 s, 55 °C for 30 s, and 72 °C for 30 s, with 30 cycles and an extension at 72 °C for 7 min. Glyceraldehyde-3-phosphate dehydrogenase (GAPDH) was used as a control, and all experiments were performed at least three times as individual repeats.

### 4.7. MTT Assay

Cell proliferation was analyzed with an MTT assay. A549 cells and H1299 cells (5 × 10^3^ cells/well) were seeded on a 96-well plate. Cells were treated with OMT (0, 100, 200 μM) for the indicated time intervals. After treatment, cells were treated with a 2 mg/mL MTT solution (30 μL/well) for 2 h and a an MTT lysis buffer (100 μL/well) was then added for overnight incubation at 37 °C. Finally, lysed MTT formazans were measured by VARIOSKAN LUX (Thermo Fisher Scientific Inc) at 570 nm. Cell viability was normalized as relative percentages in comparison with untreated controls.

### 4.8. Cell Cycle and Annexin V assays

To confirm the apoptosis, we performed cell cycle analysis. A549 cells (5 × 10^5^ cells/well) were treated with OMT alone or co-treated with paclitaxel in the indicated concentrations for 24 h. Thereafter, cell cycle and annexin V assays were performed as described previously [104].

### 4.9. Drug Combination Analyses with Paclitaxel and OMT

To confirm the combination effect between OMT (0, 50, 100, 150 μM) and paclitaxel (0, 5, 10, 15 nM), A549 cells (1 × 10^4^ cells/well) were seeded on a 96-well plate and treated with each concentration mixture for 24 h. Cytotoxicity was analyzed by MTT assay to find the optimal dose of the drugs, and cells were then evaluated with CalcuSyn (BIOSOFT, Ferguson, MO, USA) software to calculate a combination index (CI) and select a moderate combination rate. Using these data, synergy and antagonism were evaluated: CI < 1, CI = 1, and CI > 1, respectively.

### 4.10. Live and Dead Assay

A549 cells (2 × 10^4^ cells/well) were treated with OMT (100 μM) and paclitaxel (10 nM) for 24 h. Cells were stained with 5 μM Calcein AM and 5 μM Ethd-1(Ethidium homodimer-1) at 37 °C for 30 min using a Live and Dead assay (Invitrogen, Carlsbad, CA, USA). Because of the intracellular esterase activity, live cells disgregated the Calcein, so cells appeared in green. However, dead cells damaged the cellular membrane, so Ethd-1 could invade the cell and appeared in red. Stained cells were detected by an Olympus FluoView FV1000 confocal microscope (Tokyo, Japan).

### 4.11. Animals

All procedures involving animals were reviewed and approved by Kyung Hee University Institutional Animal Care and Use committee (KHUASP(SE)-17-130). Six-week-old athymic nu/nu female mice were purchased form Nara Biotec CO. (Gyeonggi-do, Korea).

### 4.12. Experimental Protocol

One week after tumor injection, tumor diameters were measured using a Digimatic caliper (Mitutoyo Company, Kawasaki, Japan). When tumors reached 0.25 cm in diameter, the mice were randomized into the following four treatment groups (*n* = 6/group). Group I, the control, was treated with PBS (100 µL; i.p.; 3 times/week), Group II was treated with OMT alone (30 mg/kg; i.p.; 3 times/week), Group III was treated with paclitaxel alone (1 mg/kg; i.p.; once a week), and Group IV was treated with both OMT (30 mg/kg; i.p.; 3 times/week) and paclitaxel (1 mg/kg; i.p.; once a week). Therapy continued for 20 days from randomization (Day 0). Tumor volumes were measured by the Digimatic caliper every 5 days, and mouse body weight was measured in 2~3 day intervals. Mice were killed 5 days after last therapy. Tumors were excised, and final weight and volume was measured using the formula V = 4/3 πr^3^. Half of the tumor tissues were fixed in formalin and embedded in paraffin for immunohistochemistry and routine hematoxylin and eosin (H&E) staining. The other half was snap-frozen in liquid nitrogen and stored at −80 °C.

### 4.13. Western Blot and Immunohistochemical Analysis of Tumor Tissues

These assays were performed as described previously [39].

### 4.14. Statistical Analysis

All numeric values are represented as the mean ± SE. Statistical significance of the data compared with the untreated control was determined using the Mann–Whitney U test. Significance was set at *p* < 0.05.

## 5. Conclusions

We report here that OP-D can potentiate paclitaxel-induced apoptotic effects, leading to the downregulation of STAT5 activation, and thus may be used in combination with chemotherapeutic agents against NSCLC.

## Figures and Tables

**Figure 1 cancers-11-00049-f001:**
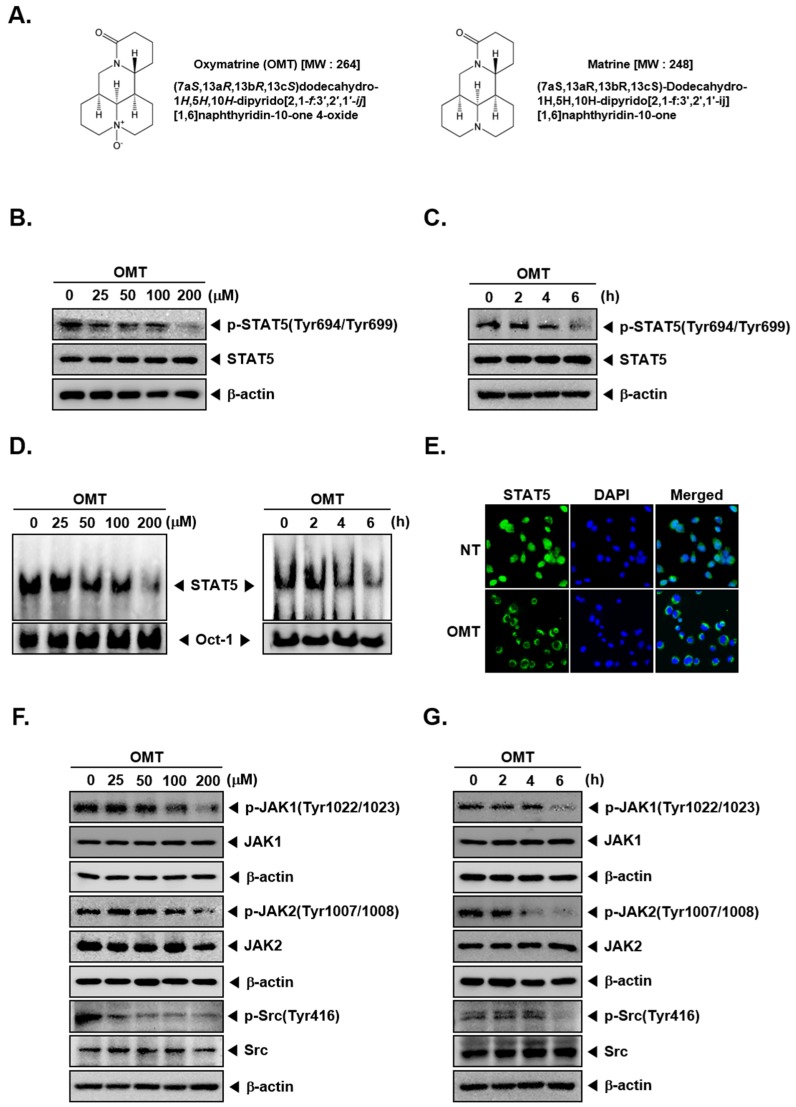
Oxymatrine (OMT) inhibits the constitutive STAT5 phosphorylation in A549 human lung cancer cells. (**A**) The chemical structure of OMT and matrine. (**B**,**C**) OMT inhibits STAT5 phosphorylation in dose- and time-dependent manners. A549 cells (5 × 10^5^ cells/well) were treated with various concentrations (0, 25, 50, 100, 200 μM) of OMT for 6 h or indicated a time interval with 200 μM OMT. Then equal amounts of whole cell lysates was analyzed via Western blot analysis and probed for phospho-STAT5 (Tyr694/Tyr699) and STAT5. (**D**) OMT inhibits the binding activity of STAT5 in A549 cells. A549 cells (5 × 10^5^ cells/well) were treated with OMT in the indicated dose- and time-dependent manners, and nuclear extracts were then analyzed by EMSA. (**E**) OMT suppressed the translocation of STAT5 to the nucleus. A549 cells (2 × 10^4^ cells/well) were treated with 200 μM OMT for 6 h, and the intracellular activity of STAT5 was then analyzed by immunocytochemistry. (**F**,**G**) OMT inhibits phosphorylation of upstream kinases. Western blot was performed as shown in Figure 1B,C, and levels of various proteins was analyzed.

**Figure 2 cancers-11-00049-f002:**
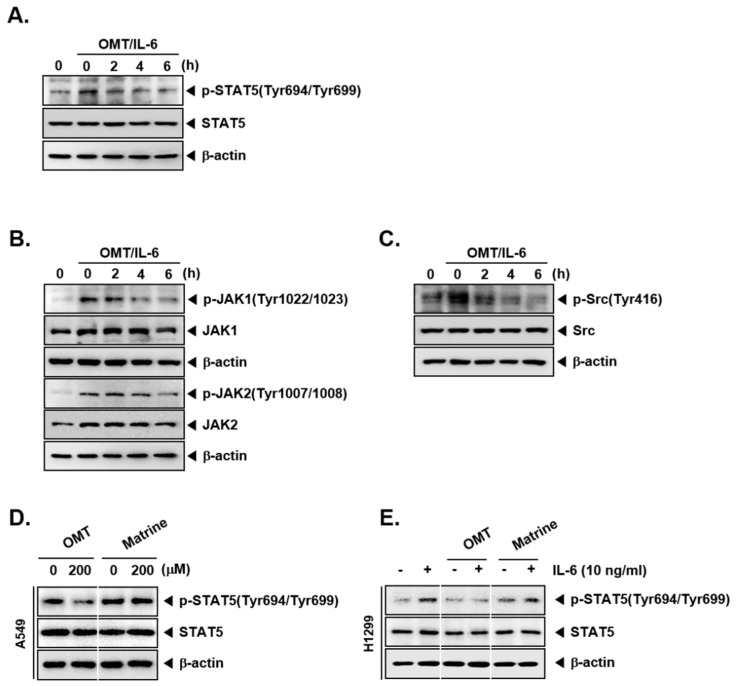
OMT can downregulate inducible STAT5 phosphorylation in H1299 cells. (**A**–**C**) H1299 cells (5 × 10^5^ cells/well) were treated with 200 μM OMT for indicated time intervals and then treated with IL-6 (10 ng/mL) for 15 min. Western blot analysis was performed as shown in Figure 1B,C. (**D**) A549 cells (5 × 10^5^ cells/well) were treated with OMT or matrine (200 μM) for 6 h. Equal amount of whole cell lysates were prepared, and Western blot analysis was performed as shown in Figure 1B,C. (**E**) H1299 cells (5 × 10^5^ cells/well) were treated with OMT or matrine (200 μM) for 6 h and then treated with IL-6 (10 ng/mL) for 15 min. Western blot analysis was performed as shown in Figure 1B,C.

**Figure 3 cancers-11-00049-f003:**
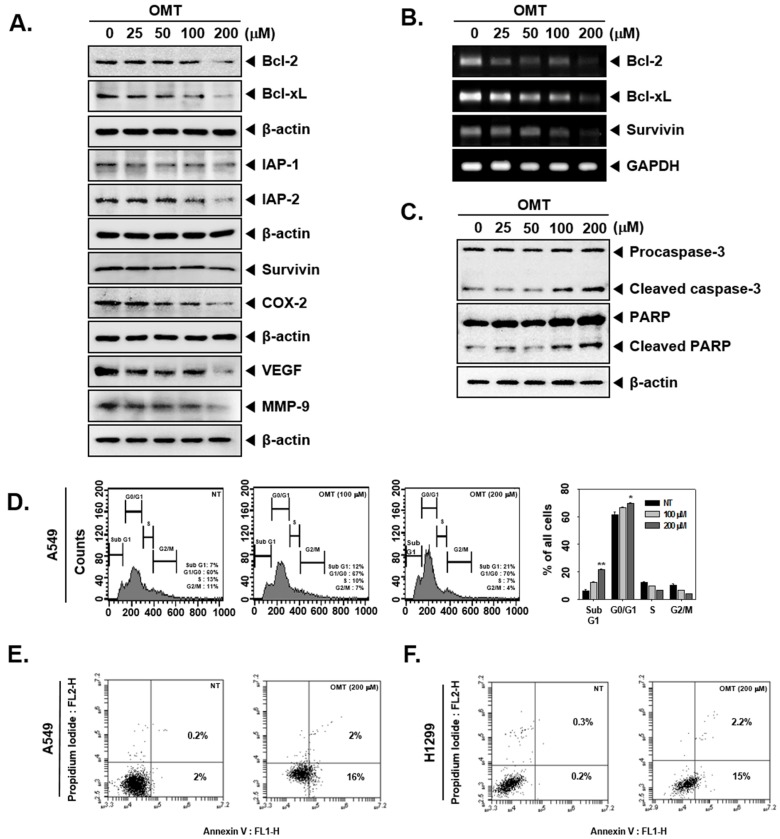
OMT promotes apoptosis in A549 cells and H1299 cells. (**A**) OMT can downregulate levels of anti-apoptotic proteins such as Bcl-2, Bcl-xl, Survivin, IAP-1, IAP-2, COX-2, VEGF, and MMP-9. A549 cells (5 × 10^5^ cells/well) were treated with OMT in indicated concentrations for 24 h. Whole cell lysates were prepared with the same amount and analyzed via Western blot analysis. (**B**) A549 cells (5 × 10^5^ cells/well) were treated with various concentrations of OMT for 24 h. Total RNA was extracted, and equal amounts were examined for expression of Bcl-2, Bcl-xl, and Survivin by RT-PCR. (**C**) Whole cell lysates were prepared with the same amount, and expression of the proteins was analyzed via Western blot analysis. (**D**) A549 cells (5 × 10^5^ cells/well) were treated with OMT (0, 100, 200 μM) for 24 h. Cells were digested with RNase A for 1 h and stained with propidium iodide, and cell cycle division was analyzed with flow cytometric analysis. The results are presented as the mean ± SE. The experiments were performed three times independently. * *p* < 0.05 and ** *p* < 0.01 compared to the control. (**E**,**F**) A549 cells (5 × 10^5^ cells/well) and H1299 cells (5 × 10^5^ cells/well) were treated with OMT (200 μM) for 24 h. After treatment, cells were stained with Annexin V-FITC and then analyzed by flow cytometry. (**G**) OMT inhibited cell growth. A549 cells (1 × 10^4^ cells/well) and H1299 cells (1 × 10^4^ cells/well) were treated with OMT (0, 100, 200 μM) for indicated time intervals. Cell viability was analyzed by MTT assay. The results are presented as the mean ± SE. The experiments were performed three times independently. *** *p* < 0.001 compared to the control.

**Figure 4 cancers-11-00049-f004:**
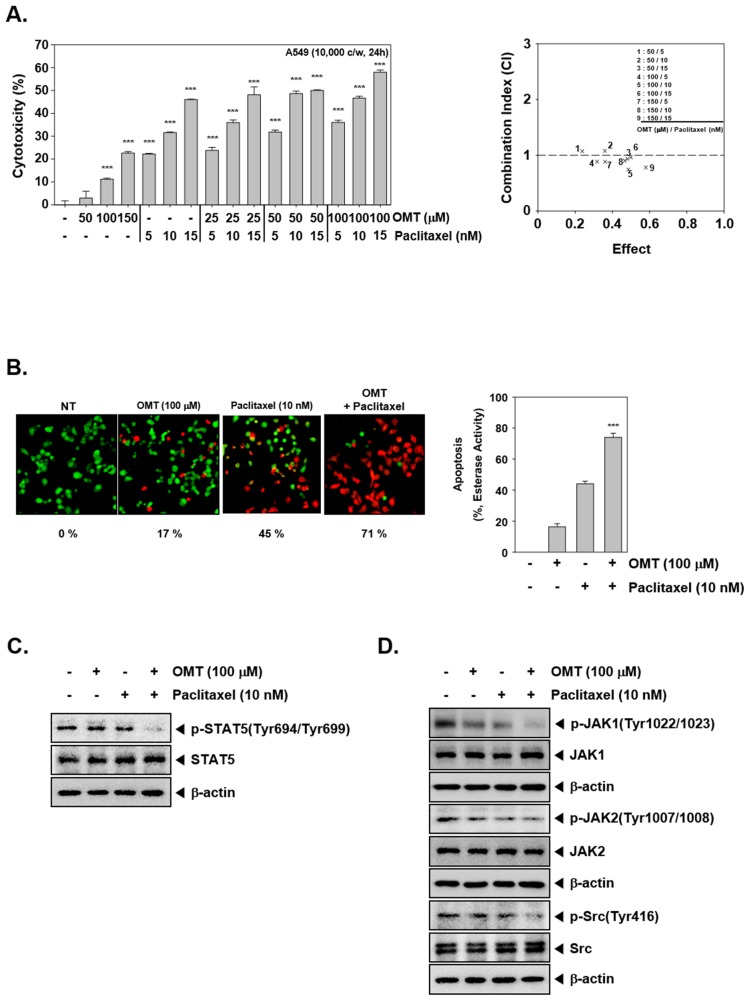
OMT and paclitaxel can enhance cytotoxicity in A549 cells. (**A**) Cytotoxicity of OMT and paclitaxel were analyzed by MTT assay. A549 cells (1 × 10^4^ cells/well) were treated with OMT and paclitaxel in various concentrations for 24 h. The average of CI values about various combination indicates that the best combination ratio was that of 100 μM OMT and 10 nM paclitaxel. (**B**) Cytotoxicity of OMT and paclitaxel were analyzed using a Live and Dead assay. A549 cells (2 × 10^4^ cells/well) were treated with 100 μM OMT and 10 nM paclitaxel for 24 h. Live cells were stained in green and dead cells were stained in red. The graph (right) shows the rate of dead cells by quantification. The results are presented as the mean ± SE. The experiments were performed three times independently. *** *p* < 0.001 compared to the control. (**C**,**D**) A549 cells (5 × 10^5^ cells/well) were treated with 100 μM OMT and 10 nM paclitaxel for 6 h. The Western blot experiments were carried out as shown above in Figure 1B,C.

**Figure 5 cancers-11-00049-f005:**
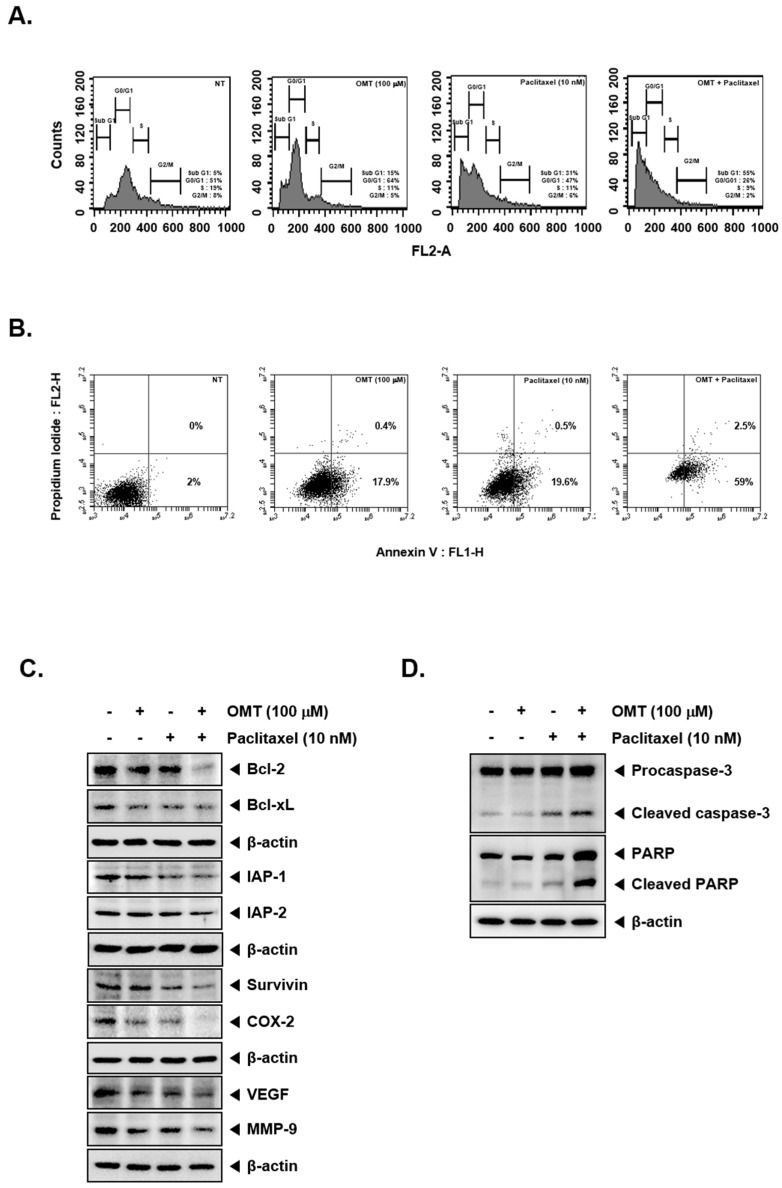
OMT and paclitaxel can induce substantial apoptosis. (**A**) A549 cells (5 × 10^5^ cells/well) were treated with OMT (100 μM) and paclitaxel (10 nM) for 24 h and then incubated with RNase A for 1 h. Cells were stained with propidium iodide, analyzed by flow cytometry. (**B**) OMT (100 μM)- and paclitaxel (10 nM)-treated A549 cells were stained with Annexin V-FITC and propidium iodide and analyzed by flow cytometry. (**C**,**D**) A549 cells (5 × 10^5^ cells/well) were treated with OMT (100 μM) and paclitaxel (10 nM) for 24 h. Equal amount of whole cell lysates were analyzed via Western blot analysis for various proteins.

**Figure 6 cancers-11-00049-f006:**
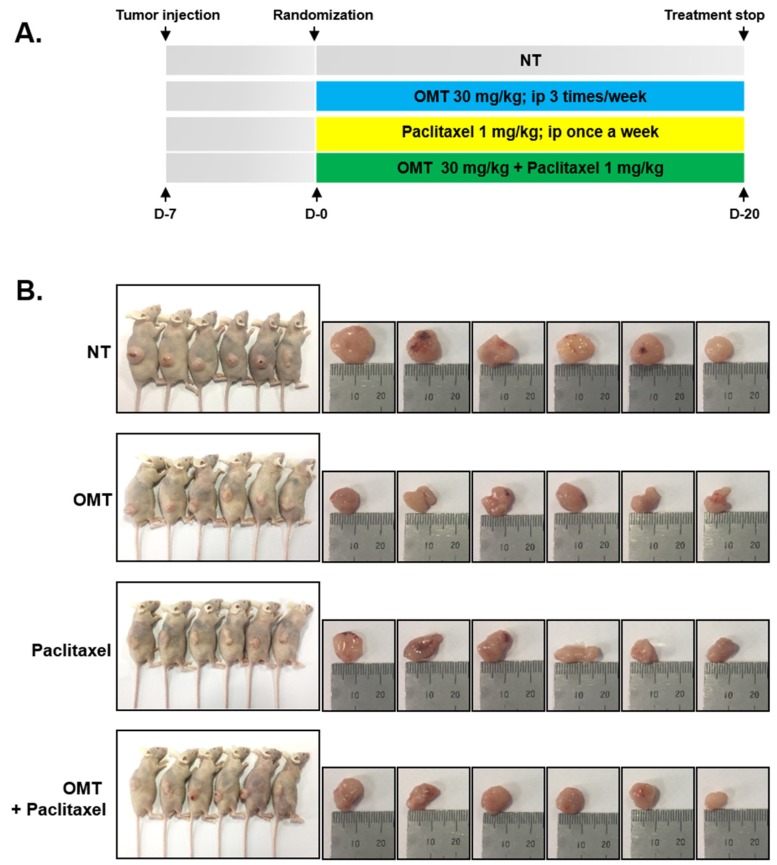
Anti-tumor effects of OMT and paclitaxel in a human lung cancer xenograft mouse model. (**A**) A549 cells (1 × 10^7^ cells/mice) were injected subcutaneously into the right flank of the mice. After 1 week of tumor cell injection, the animals were randomized into four groups. Group I was treated with PBS (100 μL; i.p.; 3 times/week), Group II was treated with OMT alone (30 mg/kg; i.p.; 3 times/week), Group III was treated with paclitaxel alone (1 mg/kg; i.p.; once a week), and Group IV was treated with a combination of OMT (30 mg/kg; i.p.; 3 times/week) and paclitaxel (1 mg/kg; i.p.; once a week) (*n* = 6). (**B**) Necropsy photographs of mice and xenograft tumors of A549 cells on Day 25. (**C**) Tumor volume was measured using Digimatic calipers on the indicated every 5 days (mean ± SE). ** *p* < 0.01 and *** *p* < 0.001 compared to the control. (**D**) Tumor weight was measured on Day 25, the last day of the experiment (mean ± SE). (**E**) Body weight was measured on the indicated days. OMT and paclitaxel combination treatment did not exhibit significant toxicity in mice (mean ± SE).

**Figure 7 cancers-11-00049-f007:**
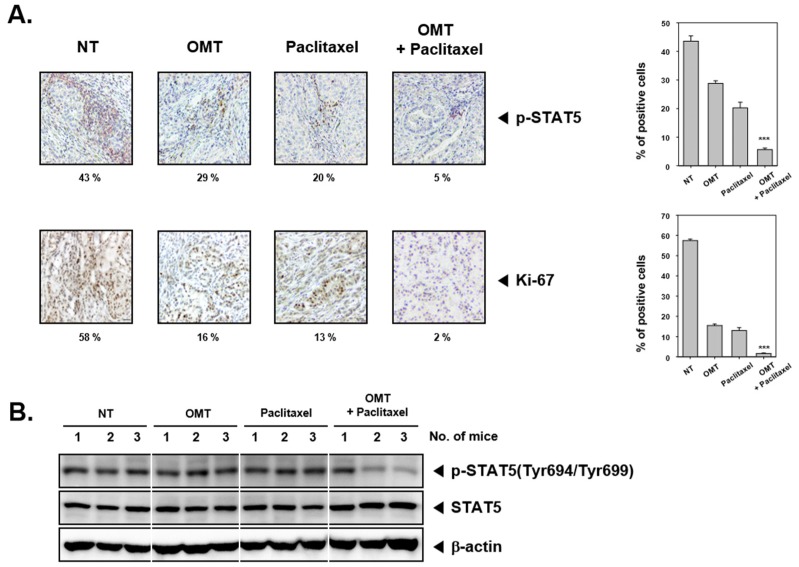
Effect of drug combination on cell proliferation, survival, and angiogenesis in lung cancer tissues. (**A**) OMT- and paclitaxel-treated mice tissues were analyzed by immunohistochemical analysis. Phospho-STAT5, and Ki-67 levels were detected in each treatment group and represented data were shown on the left panel. Solid tumors from the control and various treatment groups were fixed with 10% phosphate buffered formalin, processed, and embedded in paraffin. Sections were cut and deparaffinized in xylene and dehydrated in graded alcohol and finally hydrated in water. Antigen retrieval was performed by boiling the slide in 10 mM sodium citrate (pH 6.0) for 30 min. Immunohistochemistry was performed following manufacturer instructions (Vector Laboratories ImmPRESS™ REAGENT KIT). Quantification of phospho-STAT5 and the Ki-67 index was represented as mean ± SE on right panel. *** *p* < 0.001 compared to the control. (**B**) Western blot analysis with tissues samples showed that OMT and paclitaxel inhibited phospho-STAT5 (Tyr694/Tyr699). (**C**,**D**) Equal amount of lysates were analyzed via Western blot analysis and then probed for different proteins. The results shown are representative of two independent experiments.

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
