# Peer review of "Oxymatrine Attenuates Tumor Growth and Deactivates STAT5 Signaling in a Lung Cancer Xenograft Model"

_cancers, 2019, doi:10.3390/cancers11010049_

Reviewer 1 Report

a) The following western blot figures require individual loading control for the proteins of interest:

Fig 1 (F & G); Fig 2B; Fig 3A; Fig 4D; Fig 5C; Fig 7C

b) From Fig 6C, the difference is not visually significant between OMT & combination group and between Paclitaxel & combination group on day 25. Please justify.

Author Response

Reviewer #1

Comment 1. The following western blot figures require individual loading control for the proteins of interest:

Fig 1 (F & G) , Fig 2B, Fig 3A, Fig 4D, Fig 5C, and Fig 7C

Response: We have now inserted individual loading control in Fig 1 (F & G) , Fig 2B, Fig 3A, Fig 4D, Fig 5C, and Fig 7C.

Comment 2.  From Fig 6C, the difference is not visually significant between OMT & combination group and between Paclitaxel & combination group on day 25. Please justify.

Response: We noticed a significant decrease in both tumor size and weight in the combination group at the end of 25 days treatment (Figs.6C and 6D), without exhibiting any adverse effects.

Reviewer 2 Report

The work performed by Kwang Seok Ahn and colleagues has been well performed and is interesting for the scientific community but some details should be taken into account before its publication:

- in Abstract says “(…)various pharmacological activities We elucidated (…)” and should say “(…) various pharmacological activities. We elucidated (…)”

- in Introduction says “NSCLC is more common in women and in East Asia and is often associated (…)” and should say “NSCLC is more common in women and in East Asia is often associated (…)”

- OMT and oxymatrine are both used in the text, only one term should be used along the text. For example in page 2 row 92 says “The molecular mechanism(s) of action of oxymatrine (…)” but in same page OMT term is used.

- in page 3 rows 124 and 125 says “The result shown that OMT can indeed inhibit STAT5-DNA binding (Fig. 1D) in a dose and time-124 dependent fashion.” and should say “The result shows that OMT can indeed inhibit STAT5-DNA binding (Fig. 1D) in a dose and time-124 dependent fashion. ”

- in page 5, row 148 says “We also examined whether OMT also has modulatory effect (…)” and should say “We also examined whether OMT has modulatory effect (…)”

-in page 5 rows 150-151 says “As shown on Fig. 2A-C, OMT clearly suppressed phosphorylation of STAT5, JAK1, JAK2, and Src. ” however figures clearly indicate suppressed phosphorylation of STAT5 and Src., but it is not so clear for JAK1 and JAK2. This should be expressed in a better way.

- in page 5 row 167 says “Western blot analysis shown that OMT treated cells ” and should say “Western blot analysis shows that OMT treated cells ”.

- In Discussion, page 15 rows 319-320, says “In a similar study by the same group reported that OMT also augmented the anti-tumor activity of oxaliplatin in colon cancer cells [60]. ” and should say “A similar study reported by the same group showed thatOMT also augmented the anti-tumor activity of oxaliplatin in colon cancer cells [60]. ”

Author Response

Reviewer #2

The work performed by Kwang Seok Ahn and colleagues has been well performed and is interesting for the scientific community but some details should be taken into account before its publication:

Comment 1. in Abstract says “(…)various pharmacological activities We elucidated (…)” and should say “(…) various pharmacological activities. We elucidated (…)”

Response: We have corrected it as suggested by you. Thank you.

Comment 2.  in Introduction says “NSCLC is more common in women and in East Asia and is often associated (…)” and should say “NSCLC is more common in women and in East Asia is often associated (…)”

Response: We have corrected it. Thank you for your advice.

Comment 3. OMT and oxymatrine are both used in the text, only one term should be used along the text. For example in page 2 row 92 says “The molecular mechanism(s) of action of oxymatrine (…)” but in same page OMT term is used.

Response: We have corrected oxymatrine to OMT. Thank you.

Comment 4.  in page 3 rows 124 and 125 says “The result shown that OMT can indeed inhibit STAT5-DNA binding (Fig. 1D) in a dose and time-124 dependent fashion.” and should say “The result shows that OMT can indeed inhibit STAT5-DNA binding (Fig. 1D) in a dose and time-124 dependent fashion. ”

Response: We have modified it as suggested. Thank you.

Comment 5.  in page 5, row 148 says “We also examined whether OMT also has modulatory effect (…)” and should say “We also examined whether OMT has modulatory effect (…)”

Response: We have corrected as suggested. Thank you.

Comment 6.  in page 5 rows 150-151 says “As shown on Fig. 2A-C, OMT clearly suppressed phosphorylation of STAT5, JAK1, JAK2, and Src. ” however figures clearly indicate suppressed phosphorylation of STAT5 and Src., but it is not so clear for JAK1 and JAK2. This should be expressed in a better way.

Response: We have modified it as suggested. Thank you.

Comment 7.  in page 5 row 167 says “Western blot analysis shown that OMT treated cells ” and should say “Western blot analysis shows that OMT treated cells ”.

Response: We have modified it according to your advice. Thank you.

Comment 8.  In Discussion, page 15 rows 319-320, says “In a similar study by the same group reported that OMT also augmented the anti-tumor activity of oxaliplatin in colon cancer cells [60]. ” and should say “A similar study reported by the same group showed thatOMT also augmented the anti-tumor activity of oxaliplatin in colon cancer cells [60]. ”

Response: We have modified it as suggested. Thank you.

Reviewer 3 Report

The authors submitted a very interesting original research manuscript which deals with anti-cancer activity of alkaloid oxymatrine. The authors explored the effects of oxymatrine on STAT5 signaling cascade. As oxymatrine is a promising drug candidate with cytotoxic activity on various types of tumours, the topic of the manuscript is timely.

The authors used adequate methods. The experimental data support the results which are properly discussed. The authors cite the most important references.

I especially appreciate the patience of authors and the optimal planning of the experiments. However, I believe that the authors should take more attention to reactive oxygen species generation by this alkaloid. Oxymarine is a N-oxide which are known by their reactivity and the ROS generation. Nevertheless, the authors mention the ROS generation only briefly in Discussion when they describe the factors activating STAT5 cascade. However, oxymarine activity includes so many aspects that cannot be practically included all of them in one manuscript. From this point of view, my above-note about ROS is just a recommendation for a future investigation.

Minor issues: Figure 1A, in the chemical formula of oxymatrine, I recommend replace the double bound (N=O) by more precise bound N+—O- which better describes the reality of N-oxide chemical properties.

The name of families should be written by capital initials (line 74, Leguminosae).

Author Response

Reviewer #3

Comment 1.  The authors submitted a very interesting original research manuscript which deals with anti-cancer activity of alkaloid oxymatrine. The authors explored the effects of oxymatrine on STAT5 signaling cascade. As oxymatrine is a promising drug candidate with cytotoxic activity on various types of tumours, the topic of the manuscript is timely.

The authors used adequate methods. The experimental data support the results which are properly discussed. The authors cite the most important references.

Response: Thank you for your compliment.

Comment 2.  I especially appreciate the patience of authors and the optimal planning of the experiments. However, I believe that the authors should take more attention to reactive oxygen species generation by this alkaloid. Oxymarine is a N-oxide which are known by their reactivity and the ROS generation. Nevertheless, the authors mention the ROS generation only briefly in Discussion when they describe the factors activating STAT5 cascade. However, oxymarine activity includes so many aspects that cannot be practically included all of them in one manuscript. From this point of view, my above-note about ROS is just a recommendation for a future investigation.

Response: We thank the expert reviewer for raising an important issue. At present, it is not clear from our findings whether the STAT5 inhibitory effects of OMT are directly mediated through pro-oxidant mechanism(s). Additional studies will be carried out in future to elucidate the exact role of oxidative stress in the suppression of STAT5 phosphorylation caused by OMT in NSCLC cell lines.

Comment 3.  Minor issues: Figure 1A, in the chemical formula of oxymatrine, I recommend replace the double bound (N=O) by more precise bound N+—O- which better describes the reality of N-oxide chemical properties.

Response: We have now modified the chemical structure of oxymatrine according to your advice. Thank you.

Comment 4.  The name of families should be written by capital initials (line 74, Leguminosae).

Response: We have modified it as suggested. Thank you for your advice.

Reviewer 4 Report

The paper by Jung et al describes the effects of Oxymatrine (OMT), a component of Chinese Herbal Medicine, in lung cancer cells. The agent is cytotoxic, causes apoptosis, and affects STAT5 signaling.  OMT has activity in an in vivo mouse xenograft model.  The paper is clearly structured and reasonably well written, although rather descriptive.

Conceptual comments.

1.   OMT has been extensively investigated in a dozen or more tumor types, including lung cancer, and its mechanism of toxicity has been firmly established as apoptotic.  Extending the studies to another cell line  represents at best an incremental advance in knowledge.  I will leave it to the Editors to decide if this level of novelty is appropriate and sufficient for the Journal.

2.  The scientific rationale for selecting STAT5 as a potential mechanism is weak.  The justification is based on a common role for STAT5 signaling in cancer, but there are hundreds of such mechanisms, and it is unclear specifically why STAT5 was selected.

3.  The title suggests that STAT5 inhibition is causally responsive for OMT effects. This is not supported by experiments; no studies were performed that documented STAT5 is responsible for the observed anticancer effects.  The title needs to be changed to reflect that results are correlative.

Technical comments

1.    Use of the term “synergism” or “synergistic” to describe the interaction of OMT with Taxol  is not justified based on the data presented.

a.       Combination indices in Figure 4 are close to “1”, indicating mostly additivity.  To prove statistically significant enhancement of activity, the authors must show 1) the CI values for the combinations as avg ± SD from multiple independent experiments in Figure 4A,  2) test for statistically significant deviation from additivity by single sample t-test and 3) calculate 95% confidence intervals.  

b.       Claiming synergy by comparing non-treated vs combination-treated samples is not a proper way to measure synergism.  (Figure 4B, 6C, and 7A).  

c.       Figure 5 claims synergism but is entirely qualitative, showing data from a single experiment.

2.    Figure 4A and 4B are missing the number of independent repeats.

3.    The explanation of tissue studies was not completely clear.  Text and the legend to Figure  7 do not mention where  the material used  came from, and the Materials section is a single sentence referring to a prior publication. 

Author Response

Reviewer #4

The paper by Jung et al describes the effects of Oxymatrine (OMT), a component of Chinese Herbal Medicine, in lung cancer cells. The agent is cytotoxic, causes apoptosis, and affects STAT5 signaling. OMT has activity in an in vivo mouse xenograft model. The paper is clearly structured and reasonably well written, although rather descriptive.

Conceptual comments.

Comment 1. OMT has been extensively investigated in a dozen or more tumor types, including lung cancer, and its mechanism of toxicity has been firmly established as apoptotic. Extending the studies to another cell line represents at best an incremental advance in knowledge. I will leave it to the Editors to decide if this level of novelty is appropriate and sufficient for the Journal.

Response: Although OMT has been found to exhibit anti-tumor activity through modulation of JAK2/STAT3, EGFR/PI3K/AKT/mTOR, NF-κB, and TGFβ1/Smad signaling pathways in various tumor cells, however there are no prior reports elaborating that the anti-neoplastic effects of OMT may be primarily mediated through the attenuation of STAT5 signaling axis in NSCLC. We have also observed in our study that OMT can substantially abrogate the activation of constitutive and IL-6 induced STAT5 phosphorylation and activation of multiple upstream kinases in NSCLC cells. It was also observed to enhance the effects of paclitaxel in xenograft NSCLC mouse model through the abrogation of various biomarkers associated with tumor proliferation, survival and metastasis.

Comment 2. The scientific rationale for selecting STAT5 as a potential mechanism is weak. The justification is based on a common role for STAT5 signaling in cancer, but there are hundreds of such mechanisms, and it is unclear specifically why STAT5 was selected.

Response: We decided to choose STAT5 as it has been reported to be frequently activated by various tyrosine kinases, oxidant stress and ROS metabolism in various tumor types including NSCLC. Interestingly, in 71 NSCLC patients, immunoexpression analysis revealed significantly higher STAT5 levels in pT2 tumors and a positive correlation between STAT5 and COX-2 levels were also observed. Moreover, immunohistochemical analysis for the expression of STAT5 was carried out in 92 NSCLC samples. It was noted that STAT5 was found to be overexpressed in 41.3 % in the cytoplasm and 32.6% in the nucleus and was correlated with Bcl-xL overexpression

Hence, we decided to investigate whether the observed anticancer effects of OMT in NSCLC cells may be partially mediated through the abrogation of STAT5 signaling axis. Thanks.

Comment 3. The title suggests that STAT5 inhibition is causally responsive for OMT effects. This is not supported by experiments; no studies were performed that documented STAT5 is responsible for the observed anticancer effects. The title needs to be changed to reflect that results are correlative.

Response: We agree as due to short time provided for revision experiments, we were unable to demonstrate that the deletion of STAT5 by siRNA can reduce the observed anti-neoplastic effects of OMT in NSCLC cells. However, we will like to retain the title as it clearly summarizes the findings of our study.

Technical comments

1. Use of the term “synergism” or “synergistic” to describe the interaction of OMT with Taxol is not justified based on the data presented.

Response: We have now deleted the term “synergism” or “synergistic” from the revised manuscript. Thanks.

a. Combination indices in Figure 4 are close to “1”, indicating mostly additivity. To prove statistically significant enhancement of activity, the authors must show 1) the CI values for the combinations as avg ± SD from multiple independent experiments in Figure 4A, 2) test for statistically significant deviation from additivity by single sample t-test and 3) calculate 95% confidence intervals.

Response: We have now revised this part in the result section.

b. Claiming synergy by comparing non-treated vs combination-treated samples is not a proper way to measure synergism. (Figure 4B, 6C, and 7A).

Response: We have now deleted the term “synergism” or “synergistic” from the revised manuscript. Thanks.

c. Figure 5 claims synergism but is entirely qualitative, showing data from a single experiment.

Response: We have now deleted the term “synergism” or “synergistic” from the revised manuscript. Thanks.

2. Figure 4A and 4B are missing the number of independent repeats.

Response: The number of independent repeats has been now indicated. Thanks.

3. The explanation of tissue studies was not completely clear. Text and the legend to Figure 7 do not mention where the material used came from, and the Materials section is a single sentence referring to a prior publication.

Response: The explanation has now been added in the legend of Fig.7. Thanks.

Round  2

Reviewer 1 Report

Good job, thank you.

Author Response

Thanks a lot.

Reviewer 4 Report

see comments to Editor

Author Response

Reviewer #4

Comment 1.  In particular, the title is not supported by the data and should be changed as follows: “Oxymatrine attenuates tumor growth and deactivates STAT5 signaling in a lung cancer xenograft model”.

Response: We have now changed the title as suggested.

Comment 2.  In the legends to Figures 3 and 4 please indicate clearly how many INDEPENDENT experiments have been performed as the present formulation does not distinguish between technical or experimental triplicates.

Response: Corrected it, Thanks.

Comment 3.  Figure 3B, 6C and 7A: please provide statistical analysis also of the differences between combination and single treatments as requested (“test for statistically significant deviation from additivity by single sample t-test”), in order to address the additivity issue.

Response: Done as suggested. Thanks.